# Personal Freedom and Public Responsibility: Remaining Questions after First Case of HIV Transmission via Blood Transfusion in North Serbia

**DOI:** 10.3390/healthcare10081397

**Published:** 2022-07-27

**Authors:** Jasmina Grujić, Nevenka Bujandrić, Pavle Banović

**Affiliations:** 1Blood Transfusion Institute of Vojvodina, 21000 Novi Sad, Serbia; nevenka.bujandric@mf.uns.ac.rs; 2Department of Internal Medicine, Faculty of Medicine in Novi Sad, University of Novi Sad, 21000 Novi Sad, Serbia; 3Department of Prevention of Rabies and Other Infectious Diseases, Pasteur Institute Novi Sad, 21000 Novi Sad, Serbia; 4Department of Microbiology with Parasitology and Immunology, Faculty of Medicine in Novi Sad, University of Novi Sad, 21000 Novi Sad, Serbia

**Keywords:** blood and transfusion safety, HIV, patient blood management, blood donor ethics, transfusion-transmissible infections, pre-transfusion compatibility tests

## Abstract

The reduction in the risk of transfusion-transmissible infections (including HIV infection) is an essential part of transfusion patient care. Here, we report the first incident of HIV transmission via transfusion in north Serbia due to blood donor dishonesty, and the failure of the laboratory screen tests to detect the presence of HIV particles in his blood. Infected blood products were distributed to two recipients, and HIV infection was confirmed in one. This incident finally led to the implementation of Nucleic Acid Amplification Technology as mandatory testing of blood donors for HIV infection in Serbia and raised many questions related to the responsibility and ethics of all the participants in the blood transfusion supply chain. There is a need for the implementation of modern and non-discriminative laws in Serbia in order to reduce transfusion-transmissible infections. In addition, transfusion institutes in Serbia need to be obliged to pursue the constant upgrade of their diagnostic capacities in order to prevent similar incidents and to provide the best possible care for blood donation recipients.

## 1. Introduction

Transfusion-transmissible infection (TTI) risk reduction is an essential part of transfusion patient care. The adequate selection of blood donors (BDs) can reduce the risk of TTI up to 90% [1]. For that reason, the World Health Organization promotes and recommends blood collection from voluntary BDs that come from a population with a low risk for TTIs [2]. The risk of TTIs today remains mainly due to the existence of the infection-window period, during which no markers of infection are detected in BDs by screening tests. Technological advances have improved TTI screening for many pathogens, including the human immunodeficiency virus (HIV), where serological tests are widely replaced with nucleic acid amplification tests (NAT), reducing the window period from 56 to 12 days [1,3].

The first transmission of HIV through blood transfusion in Serbia was documented in 1985, before HIV infection screening was implemented. After that incident, a questionnaire for BDs was introduced, which included questions related to HIV-acquiring risk behavior. The system was supposed to work solely by relying on BD ethics as he/she guaranteed the truthfulness of the answers only by signing the questionnaire.

Today, Serbian National Regulations for blood donations require that BDs under risk of having TTIs are identified and excluded by the combined implementation of pre-donation selection (questionnaire), medical evaluation, and blood qualification (laboratory testing). As result of the efficient implementation of preventive measures, TTIs in Serbia rarely occur, given that more than 250,000 blood donations are collected annually. Mandatory screening of BDs for HIV exposure started in 1987, while the NAT technology was introduced almost 40 years later (i.e., 2019), after two HIV-related incidents. The first one occurred in 2002 due to laboratory staff error, where an HIV-positive donation was administered to two patients (data not published), while the second one is described and analyzed in this paper.

Here, we are reporting on HIV transmission via blood transfusion after the failure of a screen test to detect HIV infection in a BD who provided a false statement. We discuss the matter of responsibility and ethics of the participants in the transfusion supply chain, with a review of similar incidents worldwide.

## 2. Report of Incident and Aftermath

In July 2017, a thirty-year-old blood donor reported to the Blood Transfusion Institute of Vojvodina (BTIV) wishing to donate his blood. After the passing of several screening phases, he was accepted as a BD, only for it to be confirmed a few months later that he was HIV-positive (during the second visit to the BTIV). All the procedures related to this case are described according to the sequence of implementation.

### 2.1. BD Registration

When BD candidates arrive at the BTIV, they provide general information and fill in the BD questionnaire (Appendix A). After the completion and signing of the BD questionnaire, all documentation is archived under a unique bar code label. The subject described in this case denied the existence of any TTI-related risk behavior and was accepted for the next screening phase.

### 2.2. Medical Anamnesis and Physical Examination

All BD candidates are required to pass a medical examination and give responses to confidential questions related to their health and lifestyle. The purpose of the anamnestic data is to indicate the existence of possible infection/disease, TTI-related risk behavior, or TTI-related habits not specified in the BD questionnaire. In the case of any positive finding, the BD candidate is excluded from further procedures. The physical examination was focused on parameters such as blood pressure, heart rate, body mass, and body temperature, after which heart and lung auscultation and oral cavity and throat examination were performed. Finally, the existence of enlarged abdominal organs (i.e., liver and spleen) or lymph glands was checked by a medical doctor via palpation. In this case, the HIV-positive BD passed all the physical exams and did not reveal any excluding information in his anamnesis. During this procedure, the candidate was informed that BDs in Serbia do not receive any financial compensation for their donation.

### 2.3. Screening for TTI Markers

BD candidates who pass the medical examination are subjected to screening for several TTI markers. In the case of any positive finding, the BD candidate is excluded from further procedures. For the detection of HIV particles via enzyme-linked immunosorbent assay (ELISA), 6 mL of blood is collected in a vial with K2EDTA (BD, Franklin Lakes, NJ, USA; Cat. No. 367899). ELISA was performed on an EVOLIS analyzer using a Genscreen Ultra HIV Ag-Ab commercial assay (Bio-Rad, Cressier, Switzerland; Cat. No. 72388).

Additionally, 5 mL of blood was collected in an SST Advance vial (BD, USA; Cat. No. 367955) for the detection of anti-HBc, anti-HCV, and anti-Treponema pallidum IgG. Serology screening was performed via the EVOLIS analyzer using the following commercial assays: Monolisa HBs Ag Ultra (Cat No. 72348), Monolisa HCV Ag-Ab Ultra V2 (Cat. No. 72562), and Syphilis Total Ab (Cat. No. 72531). All the assays were produced by BioRad Laboratories (Hercules, CA, USA). The HIV-positive BD described in this case was negative in all the screening tests and was therefore accepted for blood donation.

### 2.4. Determination of ABO and RhD Blood Type

ABO and RhD blood type are determined in all persons finally accepted as BDs. For that purpose, 3 mL of blood was collected in a vial with K2EDTA (BD, USA; Cat. No. 367899) and analyzed via the Gel card technique using a DiaClon ABO/D+ Reverse Grouping and ABD-Confirmation assay (Bio-Rad, DiaMed GmbH, Switzerland; Cat. No. 50092 and 50041, respectively). The HIV-positive BD described here had blood type A RhD+.

### 2.5. Blood Donation Procedure

Finally, 450 mL of blood was collected from the described subject in a quadruple bag containing 63 mL of citrate-phosphate-dextrose adenine anticoagulant with a sampling system (Jiaxing Tianhe Pharmaceutical Co., Ltd., Zhejiang, China; Cat. No. OQSP-451SSBC) placed on Biomixer-323 (Ljungberg & Kögel AB, Helsingborg, Sweden; Cat. No. BM323-1). The donation process lasted 7 min, during which time the subject was constantly observed by a medical technician. No adverse reactions were recorded during or after the donation. The blood unit was finally validated for use according to the ISBT 128 Standard, while the plasma sample was archived at −80 °C, according to the BTIV standard operating procedures.

### 2.6. Detection of HIV Infection in BD

Three months later, the same person reported again to the BTIV wanting to donate blood for a second time. Following the same procedure as described previously, the person passed the questionnaire, as well as the medical exam, before his blood was screened for TTI markers. At this point, ELISA yielded a positive result for HIV infection. The result was confirmed on an Abbott Architect i2000sr analyzer using the Abbott Architect HIV I/II Ag/Ab Combo kit (Abbott, Wiesbaden, Germany, Cat. No. 2P36). The BD was informed of a possible HIV infection and was permanently excluded from making further blood donations. Simultaneously, the case of a possible HIV infection was reported to the local Center for Disease Control and Prevention.

### 2.7. Investigation and Casualties

An investigation was initiated due to the possibility that the subject was in the window period of HIV infection at the time of the first blood donation in 2017. The archived plasma sample from the first donation was forwarded to the Laboratory for Viral Diagnostics (Clinical Center of Vojvodina) where the presence of HIV RNA was confirmed via an Abbott m2000sp Molecular Real-time PCR using the HIV-1 kit (Abbott, Germany, Cat. No. 6L18). More precisely, the HIV RNA load during the time of the first blood donation was determined to be 22,836 copies/mL.

During the interview, the BD admitted that he had had sexual intercourse with other men (MSM) without the use of a condom. The identification of other components in the HIV infection chain was impossible as the BD stated that the intercourse was anonymous, and therefore, he was not able to recall the identities of his sexual partners. Additionally, he confirmed that he had purposely given a false statement on the BD questionnaire in relation to TTI-related risk activities. The BD was finally referred to the HIV Counselling Centre in the local Public Health Institute and Clinic for Infectious Diseases for further diagnostic and treatment procedures.

Further investigation revealed that components of HIV-infected blood were administered to two patients. The first patient received a platelet concentrate prepared from the HIV-positive blood donation, due to a diagnosis of systemic sepsis. The patient died shortly afterwards, and further investigation was suspended.

The second patient received suspended red blood cells from the HIV-positive blood donation as a part of the treatment for the underlying disease at the Institute of Oncology of Sremska Kamenica. A blood sample was acquired from the receiving patient, and HIV infection was detected via Siemens Advia Centaur HIV I/II Ag/Ab Combo Assay and confirmed via Inno-Lia HIV I/II Score Western Blot (Fujirebio, Tokyo, Japan, Cat. No. 80540). The test results were announced to the immediate family as the patient had died in the meantime from the underlying malignant disease.

## 3. Discussion

The described case of HIV transmission opened up many questions of an ethical and professional nature. The first question is whether the aforementioned HIV transmission would have occurred if NAT had already been implemented. If we consider 50 copies/mL as a minimum level of detection [4] and that the sample analyzed in this case had 22,836 copies/mL, we can conclude that the application of the NAT could have prevented the transmission of infection. However, testing with NAT is performed by the pooling of six samples, which means that about 67% of samples with low virus concentration can be a false negative [4]. In this particular case, given the concentration of the virus, HIV would still have been detected, even diluted in a pool.

It is important to highlight that the implementation of NAT for TTI screening in the first place was a response to a similar question raised in a German newspaper article published in 1993 [5]. The existence of nearly 1500 hemophiliacs infected with HIV through blood transfusions directed public opinion towards the conclusion that it was absurd to do nothing to reduce the risk of acquiring this TTI [5]. The subsequent investigation showed that 60% of HIV infections in transfusion recipients could have been prevented by adequate testing [4], despite the conclusion of a study published in 1993 that TTI frequency in Germany was at “low and acceptable risk” [6,7]. Responsibility for HIV transmission was finally shared by the German nation, the health insurance fund, the transfusion institutions, the Red Cross, and the hospitals, and no one was officially charged for these incidents [8]. Finally, Germany was the first country where NAT was implemented for HIV screening.

Mandatory screening with NAT for HIV was later introduced in the United States, Austria, and Japan, and in the following years, it was gradually implemented in other countries around the world [4]. It was only in 2019 that Serbia managed to provide NAT as a part of the mandatory protocol for HIV testing in BDs [9], shortly after the incident described here occurred. Implementation of NAT was probably not prioritized earlier as HIV-transmission incidents in Serbia were never the consequence of laboratory screening failure, and therefore, intervention was not considered as being of the highest priority.

Despite efforts to prevent the occurrence of TTI by NAT screening, the cases of HIV transmission by blood transfusion still occur worldwide. Two cases of HIV transmission were reported in 2017 in the USA [10], while another HIV-related incident occurred in France [10] when a BD was tested negative during his during HIV screening. Additionally, HIV was transmitted to two persons in Spain despite the BD being negative in a 44 minipool NAT [11]. A similar incident occurred in Japan, where a sample from an HIV-positive BD tested negative by NAT when diluted in the pool of 20 samples [12].

Th sensitivity and specificity of NAT in the preseroconversion infectious window period for HIV vary between the laboratories, depending on the NAT system and the protocol in use, as well as the HIV infection prevalence in the local population [13,14,15]. Therefore, in addition to laboratory screening, BD honesty and acceptance of responsibility are of pivotal importance in a TTI prevention strategy. The dishonest or misleading answers on the BD questionnaire in Serbia are most often caused by one of following reasons: (i) the persons are not aware of carrying TTI; (ii) they are seeking to be re-assured about good general health status; or (iii) they are reporting for blood donation in order to be tested for sexually transmitted diseases outside the official counselling center and to avoid stigmatization. Fearing stigma and subsequent discrimination, subjects report to donate blood and undergo testing for TTIs in a way that will allow them to avoid exposure of their lifestyles and TTI-related risk behaviors.

In response, to protect the integrity of the BD community and prevent TTIs, the Food and Drug Agency recommended screening criteria for potential BDs in the United States and raised ethical, moral, and scientific debates [16]. These screening criteria typically rely on information about the sexual behavior of persons associated with an increased risk of contracting and transmitting TTIs. Individuals identified to be at risk for TTI exposure are finally excluded from the blood donation process. The original criteria were based on the fact that AIDS was mainly diagnosed in MSM, intravenous drug users, and hemophiliacs at that time; so, in response blood operators around the world issued a deferral to these high-risk groups. Today, these criteria are being revised in the UK, the United States, Japan, Brazil, and Australia [17]. Thus, persons who declare themselves as MSM are still permanently excluded from the blood donation process in Serbia [18].

In some countries, the BD dishonesty remains the only factor that is not placed under legislative control; so, the integrity of the BD community and the blood donation recipients can be legally protected. The problem of dishonesty or misleading answers in BDs is present worldwide, given that 30% of HIV-positive BDs in Italy [19] and 50% of HIV-positive BDs in Brazil did not report any TTI-related risk factors prior to blood donation [20]. On the other hand, 100% of HIV-positive BDs registered in BTIV (north Serbia) denied having engaged in any TTI-risk behavior, including MSM. This phenomenon of dishonesty is even more concerning given that confidential medical consultation is part of each blood donation in the BTIV, where printed educational materials on the transmission of TTI are widely available.

Intentional HIV transmission is considered to be a criminal act in many countries. The USA was the first country in the world to introduce HIV-specific criminal laws, beginning in 1987, and there have been thousands of reported cases since [21]. Until 2021, there were 16 jurisdictions in the USA in criminalized behaviors with HIV, regarding donation of blood, tissues, and fluids. However, since 2014, at least nine USA states have modernized their HIV criminal laws, which include removing HIV-prevention issues from the criminal code and including them under disease control regulations [22]. Globally, most HIV criminalization prosecutions involve the severe punishment of unintentional or ‘reckless’ HIV exposure or transmission. The criminal law of Serbia contains two aspects of HIV transmission felony; “Consciously endangering another” (§ 250 para. 1), where it is punishable for knowingly endangering another, either out of negligence or with intent, and “The transmission of HIV by non-compliance with regulations and measures” (§ 250 para. 2) provides the punishment for medical workers who, within their duties, have not taken measures to prevent HIV [23]. Up to today, Serbia has not had any attempted prosecution regarding intentional HIV transmission.

Considering all the facts, the BTIV took all the measures it was legally obliged to in order to reduce the risk of transmitting HIV infection through transfusion. Even though the conditions or legal obligations for NAT testing and the process of the pathogenic inactivation of fresh frozen plasma through the law should have been introduced at the time, as in other European countries, it does not diminish the responsibility of the individuals. The use of criminal laws against people based on their HIV status can be ineffective and discriminatory and can represent a barrier to HIV prevention programs. On the other hand, the open question remains with regard to individuals who knowingly give false answers during BD screening and the consequences they should encounter, not only for HIV transmission but for other TTIs.

## 4. Conclusions

It is expected for BDs to have a strong moral obligation to avoid the harming of others whenever possible. If a person donates blood with the sole purpose of being tested for TTI, without concern for the possible repercussions, the concept of the humanity and altruism of blood donating becomes highly debatable. Therefore, there should be no place for such motives in a community of voluntary anonymous blood donors who donate a part of themselves for free in order to help a person in need. Most countries around the world are continuously introducing measures to increase the safety of blood and blood component usage. This goal can be achieved by rigorous donor selection, appropriate screening testing, and pathogenic inactivation. It is very important that the administration of one country, through its laws, supports all actions to prevent the HIV transmission by a contaminated blood transfusion. There is a need for the implementation of modern and non-discriminative laws in Serbia in order to prevent similar incidents in the future. On the other hand, the responsibility of the transfusion system is to put constant pressure on national decision makers in order achieve a constant upgrade of the diagnostic capacities, to prevent similar incidents, and to provide the best possible care for blood donation recipients.

## Data Availability

Not applicable.

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
