# Peer review of "Personal Freedom and Public Responsibility: Remaining Questions after First Case of HIV Transmission via Blood Transfusion in North Serbia"

_healthcare, 2022, doi:10.3390/healthcare10081397_

Round 1
Reviewer 1 Report
I read with great pleasure the manuscript by Grujic et al. entitled “Personal freedom and public responsibility: Remaining questions after first case of HIV transmission via blood transfusion in North Serbia”. Overall, it is a well-written paper with robust discussion about a case of HIV transmission following blood transfusion. It is well highlighted by the authors about the importance of thorough questionnaire in attempt to select the ideal blood donors. I don’t have any major comments. I only suggest removing in the abstract and conclusions the wording “Voluntary BDs can be held legally responsible if they intentionally give false information related to their activities”. As it is definitely important measure to reduce this risky behavior of not disclosing the truth, but my concern this if applied will reduce the number of blood donors’ volunteers. In addition, sometimes the BDs are not fully aware of their partners behaviors therefore we can’t blame everything on BDs. While in this case BD did not disclose his recent sexual relationship which not understandable or justified, we need to reinforce and highlight the ethical values that each and every BD should have. In my perspective, the focus should go to further stimulate the advancements of laboratory techniques (NAT/ PCR) to reduce the window period of viral infection to the shortest possible.
Author Response
Dear Editor and Reviewers:
Thank you for considering our manuscript for review and giving the opportunity for
revision. We are very grateful to the reviewers for their constructive and valuable comments which are all very helpful for revising and improving our paper.
We have incorporated most of the suggestions made by the reviewers; those changes are highlighted within the manuscript. Please see below, point-by-point response to the reviewers’ comments and concerns.
Reviewer 1
I read with great pleasure the manuscript by Grujic et al. entitled “Personal freedom and public responsibility: Remaining questions after first case of HIV transmission via blood transfusion in North Serbia”. Overall, it is a well-written paper with robust discussion about a case of HIV transmission following blood transfusion. It is well highlighted by the authors about the importance of thorough questionnaire in attempt to select the ideal blood donors. I don’t have any major comments. I only suggest removing in the abstract and conclusions the wording “Voluntary BDs can be held legally responsible if they intentionally give false information related to their activities”. As it is definitely important measure to reduce this risky behavior of not disclosing the truth, but my concern this if applied will reduce the number of blood donors’ volunteers. In addition, sometimes the BDs are not fully aware of their partners behaviors therefore we can’t blame everything on BDs. While in this case BD did not disclose his recent sexual relationship which not understandable or justified, we need to reinforce and highlight the ethical values that each and every BD should have. In my perspective, the focus should go to further stimulate the advancements of laboratory techniques (NAT/ PCR) to reduce the window period of viral infection to the shortest possible.
Thank you for your comment. We agree with you and therefore we removed suggested part of the sentence related to legal responsibility of BDs in Abstract and Conclusion sections.
We agree with your position that laboratory techniques advancement should be the backbone of TTI prevention. For that reason, we stated in conclusion:
…the responsibility of Transfusion system is to make constant pressure on national decision makers in order achieve constant upgrade of diagnostic capacities, prevent similar incidents and provide the best possible care for blood donation recipients.
Reviewer 2 Report
In the short communication by Grujic et.al describes the risks of transfusion- transmissible infections (TTI) and results of being dishonest during blood donation can increase risk of spreading HIV. In this report the authors present a patient case, which showed negative for HIV ELISA, later turned positive during the 2nd time donation. The patient dishonesty during the Blood donation has risked to spread HIV to two more patients via plasma/ blood transfusion. Reporting/Publishing this incident will increase public and health care professionals to better understand and to improve the screening criteria for blood donors and to improve the diagnostic methods for HIV or blood transfused infections in Serbia, which may reduce the risk of spreading infection via blood transfusion. However the patient sample showed a higher RNA load, but the reason for it not to be detected in ELISA remain unclear.
Author Response
In the short communication by Grujic et.al describes the risks of transfusion- transmissible infections (TTI) and results of being dishonest during blood donation can increase risk of spreading HIV. In this report the authors present a patient case, which showed negative for HIV ELISA, later turned positive during the 2nd time donation. The patient dishonesty during the Blood donation has risked to spread HIV to two more patients via plasma/ blood transfusion. Reporting/Publishing this incident will increase public and health care professionals to better understand and to improve the screening criteria for blood donors and to improve the diagnostic methods for HIV or blood transfused infections in Serbia, which may reduce the risk of spreading infection via blood transfusion. However the patient sample showed a higher RNA load, but the reason for it not to be detected in ELISA remain unclear.
Thank you for your comment.
This manuscript is a resubmission of an earlier submission. The following is a list of the peer review reports and author responses from that submission.
Round 1
Reviewer 1 Report
Dear authors,
Please, find below my comments and suggestions.
The manuscript entitled "Personal freedom and public responsibility: Remaining questions after first case of HIV transmission via blood transfusion in North Serbia" raises the issue of HIV-related incident in blood donors through a case study in the year 2017. The theme is important in public health, but in the context presented here it does not seem to me suitable to be published on Viruses, since as a pressure for health authorities, the NAT was already implemented in the country in 2018 (one year after the incident reported here).
I also highlight the fact that as part of the blood donation screening process, the donor answers questions about their health status, diet, medication use, life habits, where the candidate's sincerity is fundamental to guarantee the safety and quality of the blood. The manuscript addresses the issue of honesty, but did not explore the social characteristics that lead a person to make this decision. It is known that many donors perform this procedure in order to obtain diagnoses, exposing the system and recipients to risks, such as the presence of positive donors within the “immune window” phase. During this period, serology cannot identify the presence of anti-HIV antibodies, thus necessitating the incorporation of new, more sensitive technologies. On the other hand, the blood donation system contributes to the screening process of patients living with HIV who are not diagnosed.
In my opinion, the manuscript brings a case report that has more importance in a regional context than in a broader approach.
L51-52 - this would not be the first case of HIV transmission by blood donation, but the second after the implementation of screening for blood donors - the authors should change the title.
L59 -Please review the sentence for a better understanding.
Reviewer 2 Report
General comments: The authors describe a case of a transfusion transmitted HIV infection by a window phase infection in the donor. The donor did not declare a typical risk for HIV infection in the donor questionnaire and was thus allowed to donate blood for the first time and no testing for HIV-RNA had been performed by the blood donation service. The here presented case led to the implementation of HIV-NAT. Beside implemenation of NAT (which is already standard in many countries) the authors also demand laws which enable the prosecution of blood donors who give false statements. However, I can not fully agree with the authors concerning this point. Of course there is a responsiblity of the the donor, but the blood donation service is also responsible to use all modern tools to avoid transfusion-transmitted infection, like NAT. The authors also should consider and discuss in the text to offer blood donors the possibility of a post-donation, confidential self-exclusion (which is also standard in many countries). They also should consider and discuss the possibility that prosecution of false statements in the questionnaire might discourage people to donate blood.
Special comments: get blood donors in Serbia a financial compensation for their donation? Please add whether yes or not.
2.2. Sreening of hemoglobin... is irrelevant for the issue of thransfussion transmitted infections and can be omitted.
2.5. ABO and RhD blood group typing: is also irrelevant and can be omitted
3. Discussion: line 158 ff. HIV-NAT surely would have prevented an infection in this case, even in an minipool of 48 samples.
Line 197: person who want to be tested in the blood donation center should be offered the possibility of a confidential self exclusion. Of course it is rather speculative, but maybe, the transfusion transmitted HIV-infection reported here would have been avoided.
Reviewer 3 Report
MAJOR REMARKS.
This is a paper submitted in VIRUSES, and the conclusion are not related to science and viruses but to legal considerations that are, to my mind, highly disputable.
I do not support the views of the authors, but let the editor decide if this is pusblishable
line 43: ‘ethic’ should be replaced by ‘compliance’ to the pre-donation selection critéria’
line 138-139. ‘Identification of other components in HIV infection chain was impossible since BD stated that the intercourse was anonymous and therefore he was not able to recall the identities of his sexual partners’.
=> Does it mean that if the identity of the partner(s) was known, this(these) person(s) would have been searched for, namely in order to know if they give their blood ? That would be utterly unethical (seem my general comments on the paper).
line 201: other reasons that are mentioned in other papers are also the fear of not being able to give one’s blood (e.g. in a professional context, if several colleagues go and give their blood) or being absolutely willing to give one’s blood, or to give one’s blood seeking to be reassured about one’s global health, and not necessarily only about STIs (some BD can think they are safe regarding cancers, if their blood is accepted). The psychology of BD is more complex that the two cited situations.
line 223-236. ‘Intentional HIV transmission is considered as criminal act in many countries. The 223 USA was the first country in the world to introduce HIV-specific criminal laws, beginning 224 in 1987, and there have been thousands of reported cases since [21]. Until 2021 there were 225 16 jurisdictions in USA in criminalized behaviors in HIV regarding donation of blood, 226 tissues and fluids. However, since 2014, at least nine USA states have modernized their 227 HIV criminal laws which includes removing HIV prevention issues from the criminal 228 code and including them under disease control regulations [22]. Globally, most HIV crim-229 inalization prosecutions involve the severe punishment of unintentional, or ‘reckless’ HIV 230 exposure or transmission. The criminal law of Serbia contains 2 aspects of HIV transmis-231 sion felony; “Consciously endangering another” (§ 250 para. 1) where it is punishable for 232 knowingly endanger another, either out of negligence or with intent, and “The transmis-233 sion of HIV by non-compliance with regulations and measures” (§ 250 para. 2) provides 234 the punishment for medical workers who, within their duties, have not taken measures to 235 prevent HIV [23]. Up to today, Serbia didn’t have any attempted prosecution regarding 236 HIV transmission.
=> This is not virology, and is out of the purpose of Viruses. It is legal considerations that are highly disputable.
Lines 248-251. ‘This case points to the need for legislation so that voluntary BDs can be prosecuted for knowingly giving false information that can lead to infection of blood donation receivers. This would be an additional tool that would help reduce TTI by individuals whose reasons for donating blood are not ethical, moral and humane.
=> I do not support this conclusion at all; this is to me utterly unethical to allow condemning blood donors for allegedly putting the life of other at risk. One never know the psychology of the donors. And again, this is not virology.
lines 254-256. ‘If a person donates blood with the sole purpose of being tested for TTI without concerns of possible repercussions, the concept of humanity and altruism of blood donating becomes highly debatable’.
=> this is true and donors should be informed and educated so that they comply to the selection criteria. But legal threats are no good to me, in the frame of blood donation, as they could reduce the willingness of people to give, and threaten the blood supply.
line 263-265. ‘Our opinion is that criminal liability of the donor in relation to the pro-263 vision of false information on TTI-risk behavior should be declared as criminal felony by 264 the legislative institutions of the Serbian country.’
=> and this is not my opinion for the aforementioned reason.
This is legal matter and not virology.
The discussion should otherwise compared the situation in other countries regarding the implementation of individual NAT, and the residual risk,etc… Not the legal side.
MINOR REMARKS.
Line 34: cite proper reference (WHO)
line 39: other reference are more suitable like ref 4.
line 43: since ‘he/she’ => rephrase with ‘they’
line 47: ‘questionnaire, laboratory and medical evaluation’ => rephrase with ‘pre-donation selection (questionnaire), medical evaluation and blood qualification (laboratory testing)‘
line 51-52: cite reference related to the first incident.
lin 60: ‘After passing of several screening phases he was accepted as BD’ => what does it mean ? blood qualification was OK ? rephrase.
line 70: ‘Preliminary screening of hemoglobin and hematocrit levels’: not related with the present subject (TTIDs), should be removed
line 77: rephrase: ‘the HIV-positive BD described in this case’
line 91: ‘Screening for TTI markers’. Are the tests done before donation? Or are blood samples taken just before the donation, the donation is done and then the sample are send to the qualification lab (I supposed it’s done this way)
line 105: ‘Determination of ABO and RhD blood type’: not related with the present subject (TTIDs), should be removed
line 160: 50 copies/ml is the minimum limit of detection: it depends on the technics. The current techniques used (Ultrio, Ultrio Plus, cobas MPX) are lower.
line 163: ‘that about 67% of samples with low virus concentration can be false negative’. not really precised as no viral load is mentioned here, and is probably related to the context in the cited reference. Should be rephrased.
line 186-7: ‘HIV-related incident occurred in France 186 [10] when BD was tested negative during his during HIV screening’. Not exactly, correct: no indicent was described as no transmission to the blood recipients was shown. It was a pre-ramp-up phase gone undetected by NAT.
line 197: ‘i) person is not aware that he/she is carrying TTI’. So how can he/she ‘lie’ ? Not knowing one’s status or risk factor does not imply that one is ‘lying’.
line 204: REF 16 is incorrect. It was published in 1997 and does not correspond to a WHO recommandaton.
line 208: ‘that at the time’ => rephrase ‘at that time’, and precise which period is referred to. Probably related to the time of REF 16, but this paragraph is really unclear.
Line 210: In Greece and France, the sexual orientation is not a selection criterion anymore.
line 213: rephrase: ‘In some countries, THE BD dishonesty remains THE only factor that is not placed under legislative control, so THE integrity of THE BD community and blood donation receivers RECIPIENTS can be legally protected.
+ unsure what it means as the blood donors community and the recipient community are not the same legal categories.
